# Differences between Healthy-Weight and Overweight Serbian Preschool Children in Motor and Cognitive Abilities

**DOI:** 10.3390/ijerph191811325

**Published:** 2022-09-09

**Authors:** Boris Banjevic, Dragana Aleksic, Aleksandra Aleksic Veljkovic, Borko Katanic, Bojan Masanovic

**Affiliations:** 1Faculty for Sport and Physical Education, University of Montenegro, 81400 Niksic, Montenegro; 2Faculty for Sport and Physical Education, University of Prishtina, 38218 Leposavic, Serbia; 3Faculty of Sports and Physical Education, University of Nis, 18000 Nis, Serbia

**Keywords:** preschoolers, obesity, motor abilities, cognitive functions, BMI, TGMD-2, BOT-2

## Abstract

The aim of this study was to determine the differences between healthy-weight and overweight 5–6-year-old preschool children in fine and gross motor skills and cognitive abilities. There were 91 subjects, preschool children (41 boys and 50 girls), who participated in this cross-sectional study. The body mass index (BMI) was calculated based on measures of body height and body mass, and WHO cutoff points were used for the assessment of the children’s nutrition status. Fine motor abilities were determined using two Bruininks–Oseretsky (BOT-2) subtests, and gross motor skills are determined by the Test of Gross Motor Development (TGMD-2), while cognitive abilities were tested by the School Maturity Test (TZŠ+). Based on an independent-samples *t*-test, a difference in two out of three variables of gross motor skills was determined: manipulative skills and total gross motor skills between healthy-weight and overweight children, while in fine motor abilities and cognitive abilities there was no difference between these two groups. Although significant differences were found only in gross motor skills between healthy and overweight preschool children but not in fine motor skills and cognitive abilities, further longitudinal studies are required to understand the mechanisms of this, including the possible role of psychological factors.

## 1. Introduction

Obesity is becoming more and more present among children, and currently it is one of the major public health problems [1]. Obesity among children is connected with various health problems such as diabetes, asthma, hypertension, atherosclerosis, and psychosocial problems [2]. According to the data by Spiotta and Luma [3], today almost a third (31%) of children in the world have a body mass above the healthy levels, which is quite worrying.

The high body mass and overweight of preschool children became a bigger global concern [4,5], especially because it is well-known that early childhood is a critical factor for predicting obesity in adolescence [6,7]. The body mass index (BMI) in the preschool period is connected to biomarkers of the risk of cardiovascular diseases [8] and higher BMI in childhood and later on in adolescence, which increases health problems [6,9].

Researchers indicate that a high percentage of body mass and obesity have a negative effect on motor performance [10,11]. Recent studies show that a higher BMI is negatively connected to basic motor capabilities [12,13]. It is considered that the development of gross motor skills in children with excess weight proceeds more slowly compared to healthy-weight children [14]. One of the problems is that children who are overweight tend to avoid physical activities, which may result in a further decline in motor capabilities as well as increasing their body mass even more [10].

Some authors point out a negative connection to cognitive functions [15] and growing evidence of a possible association between overweight and poor cognitive function. Previous studies have shown a relation between overweight and intelligence quotient or poor general cognitive performance as well as relations with a variety of cognitive domains, including executive function, memory, verbal and motor abilities, and attention, at different ages [16,17]. Guxens et al. [17] found that higher cognitive function at 4 years, specifically executive function and verbal skills, was associated with a decreased risk of being overweight at 6 years. 

Developmental coordination disorder (DCD) is a condition in which gross and fine motor skills are impaired. DCD negatively correlates with body composition and the level of physical activity [18,19] as well as numerous developmental problems, such as learning, reading [20], behavior, and speech [21]. Dewey et al. [20] point out that children with DCD, irrespective of the extent of the problem, experience learning disabilities, attention deficit, and limitations in psychological functioning in general. That is why it is very important to note any irregularities in motor coordination and to correct them by carrying out adequate physical activity programs [22].

It is believed that difficulties in performing movement skills in overweight children extend to the performance of fine motor tasks, which indicates that the problems are not exclusively related to the movement of excess mass in relation to gravity. Instead, it suggests that obese children also have difficulties with the cognitive processes they need to plan motor activities [10,11].

On the basis of the given information regarding obesity in childhood and its connection with difficulties with motor skills and cognitive abilities, an insight is gained about the importance of dealing with this topic [23], especially when it is known that motor and cognitive development are related [24] and both domains of development should be examined. This is the first study to evaluate motor and cognitive development in overweight preschoolers as compared to healthy-weight peers in Serbia. In this regard, the goal of this research was to determine the differences between healthy-weight and overweight 5–6-year-old Serbian preschool children in fine and gross motor skills and cognitive abilities.

## 2. Materials and Methods

### 2.1. Participants

A total of 91 children (age range 5–6 years) participated in this cross-sectional study. The sample of participants was taken by random sampling from the “Ljubica Vrebalov” Preschool in Požarevac (Serbia). The criterion for the inclusion and selection of subjects was the following: healthy children (children without any disease or disorders) of both sexes, aged five to six years, who were not involved in any sport except the physical activities included in the kindergarten curriculum.

### 2.2. Anthropometric Characteristics

Standardized anthropometric instruments were used to measure anthropometric characteristics. The measurement was performed according to an established international biological procedure [25]. The body mass index was calculated based on the standard formula: BMI = BM (kg)/BH (m)^2^ (BM—body mass, BH—body height). The body mass index has a high correlation with the amount of body fat and for these reasons is used as an indicator of nutritional levels in children [26].

The children were classified, according to BMI status, into two groups: healthy weight and overweight (Table 1). The distribution according to BMI status was made according to reference values of the World Health Organization for the body mass index for age and sex for children [27].

### 2.3. Fine Motor Skills Assessment

Subtests from the BOT-2 (Bruininks–Oseretsky Test of Motor Proficiency) battery were used to assess fine motor skills. The BOT-2 tests have been validated for age-specific preschool children [28]. For the aims of this study, two subtests were used: fine motor integration (assessed using eight tasks) and manual dexterity (five tasks), which together compose the total fine motor skills, which represents the third variable that will be discussed below.

### 2.4. Gross Motor Skills Assessment

The TGMD-2 (Test of Gross Motor Development) was used to assess gross motor abilities. In this test, the motor skills of children are evaluated based on the observational technique. The TGMD-2 test has also been validated for a given preschool age [29]. The TGMD-2 contains 12 motor tasks divided into two subtests, which together provide a total gross motor composite, and that composite represents the third variable that will be discussed below.

### 2.5. Cognitive Abilities Assessment

Three subtests of the Maturity Test for School (TZŠ+) were used to assess the cognitive abilities of children: visual memory, the stacking of cubes, and code. The research results showed high validity and reliability of the Maturity Test for School and suggest that TZŠ+ has very high correlations with the cognitive tests TIP-1 and Raven’s color matrix [30]. Visual memory is a test designed to assess the ability to remember and pay attention, consisting of 15 tasks. Stacking cubes is a test designed to assess the abilities of visual–motor coordination, perceptual organization, and planning, containing eight tasks. The code is a test that assesses the ability to learn from experience, concentration, and visual–motor coordination, and it contains 25 tasks. Together, these three subtests give the total cognitive abilities composite: the fourth variable.

### 2.6. Variables

For each subtest, the total score was measured, and the obtained value was then converted according to the standardized BOT-2, TGMD-2, and TZŠ+ tables in relation to the gender and age of the examinees, as prescribed by the authors of these tests [29,31,32]. Therefore, the standardized data were entered for further processing, and there were three variables of fine motor skills, three variables of gross motor skills, and four variables of cognitive abilities.

### 2.7. Measurement Organization

The testing of all children in this cross-sectional study was carried out in the hall of the “Ljubica Vrebalov” Preschool in Požarevac. The tests were performed at the same time (11 AM) in order to exclude daily measurement variations. The air temperature in the room during testing ranged from 22 °C to 26 °C.

### 2.8. Statistics

Descriptive analysis was used for the distribution of respondents’ data, including mean values and standard deviations. Independent-samples *t*-tests were used to determine differences between groups of healthy-weight and overweight preschool children. The significance of the conclusion was established at the level of *p* < 0.05. The effect size was estimated using Cohen’s d effect size. The criteria for determining the size of the impact were: <0.20 trivial (t); 0.20–0.50 small (m); 0.50–0.80 moderate (in); 0.80–1.3 large (v), and >1.3 very large (vv) [33]). The statistical program SPSS 26 was used for data processing (Statistical Package for Social Sciences, v26.0, SPSS Inc., Chicago, IL, USA).

## 3. Results

In Table 1, it is noticeable that 72.5% of the children were healthy weights, while 27.5% of children were overweight. A similar distribution of healthy-weight and overweight children was observed in the groups of boys (73.2% vs. 26.8%) and girls (72% vs. 28%). Table 2 shows the differences between healthy-weight and overweight children in fine motor, and gross motor, and cognitive abilities. When it comes to anthropometric characteristics, a significant difference was found in body weight (t = −7.251, *p* = 0.001) and (t = −9.574, *p* = 0.001).

Based on the independent-samples *t*-test, a significant difference was found in two of the three variables of gross motor skills, namely, manipulative skills (t = 3.033, *p* = 0.003) and total gross motor skills (t = 2.820, *p* = 0.006) between healthy-weight and overweight children, while in fine motor skills and cognitive abilities, there was no significant difference between the groups. Based on Cohen’s criterion [33], it was determined that there was a large effect size in body weight and BMI and medium effect sizes in manipulative and gross motor skills (0.78 and 0.71, respectively).

## 4. Discussion

The aim of this research was to examine the differences in fine and gross motor and cognitive abilities between healthy and overweight children in Serbia. The results confirmed statistically significant differences in gross motor skills but not in fine motor and cognitive abilities. However, the mean values suggested that healthy-weight children achieved better results in almost every subtest. 

The results of our study show differences in gross motor skills, which is in line with Krombholz (2013), who reported that overweight children had inferior gross motor skills compared with normal-weight children in a sample of German preschoolers. They are also in line also with Gentier et al. [11], which supported the fact that excess mass interferes with optimal performance in overweight children. The authors of this study stated that overweight children improved in manipulative skills performance as they matured, whereas the healthy-weight children group maintained a high-performance level regardless of age. In this study, the difference in the performance of gross motor skills had a large magnitude, especially because of the difference caused by manipulative skills performance. This information is very important because an earlier study also confirmed that manipulative skills in childhood were associated with time in organized activity in adolescence [34]. The ability to perform object control skills (such as catching, throwing, and kicking) competently in childhood may be a significant factor in subsequent engagement in adolescent physical activity. 

Our results did not identify differences in fine motor skills, contrary to earlier studies where overweight children achieved lower scores on fine motor tasks [10,11] and in which the authors supported the idea that childhood obesity is detrimental for manual dexterity. In these studies, fine motor accuracy tests showed trends in BMI-related differences, suggesting that overweight children also perform worse than healthy-weight preschoolers when precision is required for fine motor tasks. This is not in line with our results, but our participants were younger, so there were no significant differences in the fine motor skills of healthy and overweight preschool children. Castetbon and Andreyeva [35] found that there was no significant association between obesity and fine motor development in a population of children aged 4 to 6 years, although this association persisted at later ages. These insufficient and conflicting data indicated the need for more complete research in this area. It should be emphasized that fine motor performance is not directly affected by the amount of excess mass that participates in movement, so it is not enough to explain the differences between BMI groups. That is why some authors suggest that there could be a deficit in the integration and processing of sensory information in overweight children [36], and our investigation also included an investigation of differences in cognitive abilities. 

This study did not confirm differences in cognitive abilities between the two groups of children, and this is in line with previous studies where either no differences between weight groups were found or overweight children scored less in cognitive tests [12,37,38,39,40]. Krombholz [40] suggested that the association between weight and cognitive performance is not clear because some studies have found lower mental abilities in overweight compared to healthy-weight children, while other studies have found no evidence for such an association. A longitudinal investigation demonstrated that early motor difficulties in preschool children have considerable effects on the outcomes of academic achievement and psychosocial maladaptation up until the sixth grade [41] and that higher general cognitive abilities at age 4, particularly executive function and verbal, quantitative, and memory skill scores, were associated with a lower likelihood of being overweight at age 6, after adjustment for a large list of covariates, including socioeconomic factors and maternal BMI [17]. 

This study also has some limitations, two of which are the most important. First, in further studies, the respondents should be divided according to BMI status into several groups, primarily separating the overweight children from the obese children. Second, research should be conducted on a larger sample with the inclusion of children from different parts of Serbia, not just from one region.

The strengths of this study include a sample from randomly selected Serbian preschool children and the fact that we included three areas of motor development. It is recommended that future research also involve motor abilities as well as the emotional development of children. 

Based on this study, the importance of engaging in physical activity, especially at an early age, should be pointed out. Therefore, kindergartens should follow the guidelines of the WHO Global Strategy on Diet, Physical Activity, and Health [42]. The practical implications for preschool children relate to the design of a physical exercise program that will affect fine and gross motor skills and, at the same time, include solving certain cognitive tasks.

## 5. Conclusions

Significant differences were found only in gross motor skills between healthy-weight and overweight preschool children but not in fine motor skills and cognitive abilities. However, considering the small sample size and that only one age group of preschoolers was included, further longitudinal studies are required to understand the mechanisms of this problem, including the possible role of psychological factors.

## Figures and Tables

**Table 1 ijerph-19-11325-t001:** Distribution of the sample according to gender and BMI.

	Boys (41) (45.1%)	Girls (50) (54.9)	Total (91) (100%)
Healthy weight	30 (73.2%)	36 (72.0%)	66 (72.5%)
Overweight	11 (26.8%)	14 (28.0%)	25 (27.5%)

**Table 2 ijerph-19-11325-t002:** Normal and Overweight Children differ in Motor and Cognitive Skills.

	BMI	Mean	St. D.	t	*p*	Cohen’s d
Age	Healthy weight	6.05	0.37	−1.644	0.104	0.38
	Overweight	6.20	0.41			
Body height (cm)	Healthy weight	117.83	5.26	−1.689	0.095	0.37
	Overweight	120.15	7.17			
Body weight (kg)	Healthy weight	21.31	2.42	−7.251	0.000 *	1.41
	Overweight	27.22	5.37			
BMI	Healthy weight	15.32	1.06	−9.574	0.000 *	1.88
	Overweight	18.69	2.30			
Fine motor integration	Healthy weight	13.42	4.04	−0.058	0.954	0.01
	Overweight	13.48	4.23			
Manual dexterity	Healthy weight	12.35	4.19	1.090	0.279	0.27
	Overweight	11.28	4.14			
Fine motor skills	Healthy weight	25.77	6.18	0.671	0.504	0.15
	Overweight	24.76	7.04			
Locomotor skills	Healthy weight	7.00	1.65	1.920	0.058	0.46
	Overweight	6.28	1.46			
Manipulative skills	Healthy weight	7.56	1.77	3.033	0.003 *	0.78
	Overweight	6.40	1.15			
Gross motor skills	Healthy weight	14.56	3.07	2.820	0.006 *	0.71
	Overweight	12.68	2.10			
Visual memory	Healthy weight	3.12	0.54	−1.552	0.124	0.36
	Overweight	3.32	0.56			
Stacking cubes	Healthy weight	4.27	0.83	0.976	0.332	0.22
	Overweight	4.08	0.86			
Code	Healthy weight	3.32	0.79	1.674	0.098	0.39
	Overweight	3.00	0.87			
Total cognitive abilities	Healthy weight	3.57	0.55	0.828	0.410	0.19
	Overweight	3.47	0.50			

* statistical significance.

## Data Availability

Data will be provided to all interested parties upon reasonable request.

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
