# Peer review of "Differences between Healthy-Weight and Overweight Serbian Preschool Children in Motor and Cognitive Abilities"

_ijerph, 2022, doi:10.3390/ijerph191811325_

Round 1
Reviewer 1 Report
Differences between Healthy weight and Overweight Preschool Children in Motor and Cognitive Abilities
Citations
They are not in the correct format (e.g [1] etc..)
Sub-headings
Should be: 2. Material and Methods / 2.1 Participants
Abstract
P values not normally included in abstract.
Introduction
The evidence that increased weight impacts on motor tasks is repeated too many times. However, cognitive impairments relating to weight are not explained.
Method
Re-name ‘subjects’ as participants or volunteers. What does ‘healthy children’ mean? Does this refer to the BMI?
Results
P values should be expressed as p<.001 (when .000). Text should include the df and T values.
Overall
This is an interesting paper and the authors acknowledge the limitations in sample numbers. However, they do not talk about conditions such as Developmental Coordination Disorder (sometimes called Dyspraxia) where gross and fine motor skills are impaired along with executive functioning skills.
The children described here must be described in more detail – eg. are any of the children neurodiverse?
Author Response
Reviwer 1
Differences between Healthy weight and Overweight Preschool Children in Motor and Cognitive Abilities
Citations
They are not in the correct format (e.g [1] etc..)
We corrected them
Sub-headings
Should be: 2. Material and Methods / 2.1 Participants
We added sub-headings
Abstract
P values not normally included in abstract.
We removed them
Introduction
The evidence that increased weight impacts on motor tasks is repeated too many times. However, cognitive impairments relating to weight are not explained.
Thank you for the suggestions. We added one paragraph about correlation between cognitive ability and weight status (Page 2, Paragraph 1).
Method
Re-name ‘subjects’ as participants or volunteers. What does ‘healthy children’ mean? Does this refer to the BMI?
Renamed in participants.
‘Healthy weight’ means normal weight. The distribution according to BMI status was made according to reference values of the World Health Organization for body mass index for age and sex by children (Onis et al., 2007).
‘Healthy children’ in one sentence mean children without any disease or disorders.
Results
P values should be expressed as p<.001 (when .000). Text should include the df and T values.
We corrected p values and added t values in the text.
Overall
This is an interesting paper and the authors acknowledge the limitations in sample numbers. However, they do not talk about conditions such as Developmental Coordination Disorder (sometimes called Dyspraxia) where gross and fine motor skills are impaired along with executive functioning skills.
Thank you. We added paragraph about DCD (Page 2, Paragraph 2).
The children described here must be described in more detail – eg. are any of the children neurodiverse?
We added describe in the text: Healthy children (children without any disease or disorders) of both sexes, aged five to six years, who were not involved in any sport, except physical activities included in the kindergarten curriculum
Reviewer 2 Report
The authors investigate the differences between healthy weight and overweight preschool children in Serbia regarding fine and gross motor skills and cognitive abilities.
The research design is appropriate, and the authors acknowledge its limitations. The article is relevant and timely. The paper is straightforward and well-written. Notwithstanding this, I include several minor comments and suggestions which the authors can address in order to improve the paper.
· Title: I would suggest including that the study was conducted in Serbia.
· Lines 59-70: The text reads “who were not involved in any form of organized physical exercise”. What does this mean exactly? I would suggest clarifying this.
· Lines 79-80: The text reads “The children were classified according to BMI status into two groups: healthy weight, 80 and overweight”. How about underweight subjects? Are there any? Were they removed from the study?
· I would suggest including some policy implications based on the results.
Author Response
Reviewer 2
The authors investigate the differences between healthy weight and overweight preschool children in Serbia regarding fine and gross motor skills and cognitive abilities.
The research design is appropriate, and the authors acknowledge its limitations. The article is relevant and timely. The paper is straightforward and well-written. Notwithstanding this, I include several minor comments and suggestions which the authors can address in order to improve the paper.
- Title: I would suggest including that the study was conducted in Serbia.
We changed the title.
- Lines 59-70: The text reads “who were not involved in any form of organized physical exercise”. What does this mean exactly? I would suggest clarifying this.
Changed to: ….children who were not involved in any sport, except physical activities included in the kindergarten curriculum.
- Lines 79-80: The text reads “The children were classified according to BMI status into two groups: healthy weight, 80 and overweight”. How about underweight subjects? Are there any? Were they removed from the study?
We had only two children with border weight, and they are in healthy weight group. There were not underweight subjects among them.
I would suggest including some policy implications based on the results.
We added paragraph (Page 6, Paragraph 3).